# Taurochenodeoxycholic Acid Increases cAMP Content via Specially Interacting with Bile Acid Receptor TGR5

**DOI:** 10.3390/molecules26237066

**Published:** 2021-11-23

**Authors:** Youchao Qi, Linkai Shi, Guozhen Duan, Yonggui Ma, Peifeng Li

**Affiliations:** 1Department of Basic Veterinary Medicine, College of Veterinary Medicine, Inner Mongolia Agricultural University, Hohhot 010010, China; yaolixueyouchao@163.com (Y.Q.); SLK18797328002@163.com (L.S.); 2Key Laboratory of Clinical Diagnosis and Treatment Techniques for Animal Disease, Ministry of Agriculture, Hohhot 010010, China; 3Department of Veterinary Medicine, College of Agriculture and Animal Husbandry, Qinghai University, Xining 810016, China; 4Agricultural and Pastoral Bureau of Togtoh County, Togtoh 010200, China; 5Academy of Agriculture and Forestry Sciences, Qinghai University, Xining 810016, China; 2020990013@qhu.edu.cn; 6Key Laboratory of Medicinal Animal and Plant Resources of Qinghai Tibetan Plateau, Qinghai Normal University, Xining 810008, China; 7Academy of Plateau Science and Sustainability, Qinghai Normal University, Xining 810008, China

**Keywords:** 239T cells, TGR5, taurochenodeoxycholic acid, cyclic adenosine monophosphate, interaction

## Abstract

Taurochenodeoxycholic acid (TCDCA) is one of the main components of bile acids (BAs). TCDCA has been reported as a signaling molecule, exerting anti-inflammatory and immunomodulatory functions. However, it is not well known whether those effects are mediated by TGR5. This study aimed to elucidate the interaction between TCDCA and TGR5. To achieve this aim, first, the TGR5 eukaryotic vector was constructed. The expression level of TGR5 in 293T cells was determined by immunofluorescence, real-time quantitative PCR (RT-PCR, qPCR), and Western blot. The luciferase assay, fluorescence microscopy, and enzyme-linked immunosorbent assay (ELISA) were recruited to check the interaction of TCDCA with TGR5. TCDCA treatment in 293T cells resulted in TGR5 internalization coupled with a significant increase in cAMP luciferase expression. Our results demonstrated that TCDCA was able to bind to the TGR5 receptor and activate it. These results provide an excellent potential therapeutic target for TCDCA research. Moreover, these findings also provide theoretical evidence for further TCDCA research.

## 1. Introduction

BAs are major components of bile and can be divided into free bile acids and conjugated bile acids based on their chemical structure. Free bile acids include cholic acid (CA), deoxycholic acid (DCA), chenodeoxycholic acid (CDCA), and lithocholic acid (LCA). Conjugated bile acids include glycocholic acid (GCA), taurocholic acids (TCA), glycochenodeoxycholic acid (GCDCA), and TCDCA [1]. The main physiological function of BAs is to act as a surfactant to emulsify dietary fats and lipids into micelles and to promote the absorption of lipids and the fat-soluble vitamins A, D, E, and K from the intestine, thus facilitating digestion [2,3]. BAs are also amphipathic molecular, meaning that they can prevent precipitation of cholesterol crystals, decreasing cholesterol stone formation [4,5]. Moreover, BAs have another function in anti-inflammation and immunology. TCA promotes interleukin 1 (IL-1) and interleukin 6 (IL-6) formation in the serum, strengthening humoral immunity via the inhibition of the phagocytosis of the monocyte macrophages [6]. Current evidence suggests that ursodeoxycholic acid (UDCA) can provide multiple benefits, including the decreased hydrophobicity of the bile acid pool, increased hepatobiliary secretion, reduced inflammation, and cell death [7]. Meanwhile, in 2011, Yusuke Iguchi and colleagues reported that UDCA activated TGR5, providing significant evidence and a method to continue the research on UDCA and other BAs [8].

TCDCA, as a member of BAs, enhances the apoptotic rate in NR8383 cells and inhibits the expression of the IL-1 and tumor necrosis factor α (TNF-α) via the protein kinase C (PKC)-c-Jun N-terminal kinase (JNK)/P38-P53 signal pathway [9]. It was reported that BAs significantly increase the proliferation of HepG2 cell lines and that TCDCA down-regulates the expression of a tumor suppressor gene, the CCAAT enhancer-binding protein α (CEBPα), in HepG2 cell lines [10]. So far, no evidence has shown that TCDCA can exert these functions through the TGR5 receptor. The G protein-coupled receptor (GRCR) TGR5 is a bile acid-activated membrane receptor [11]. The compound 6α-ethyl-23 (*S*)-methyl-cholic acid (EMCA, INT-777), which has a similar structure to TCDCA, activates TGR5 and stimulates adenylate cyclase (AC), intracellular cAMP production, and protein kinase A (PKA) activation [12]. Therefore, our main aim is to elucidate whether there is an interaction between TCDCA and the TGR5 receptor and, importantly, to provide a method that can be used to research ligands and receptors as well as to produce experimental evidence that furthers the further research on TCDCA.

## 2. Materials and Methods

### 2.1. Chemicals and Reagents

TCDCA (purity ≥ 98%) and TLCA (purity ≥ 97%) were purchased from Sigma Chemical Co. (St. Louis, MO, USA). INT-777 (purity ≥ 98%) was purchased from MedchemExpress Co. (Princeton, NJ, USA). Sodium dodecyl-sulfate polyacrylamide gel electrophoresis (SDS-PAGE) was purchased from Beyotime Co. (Beijing, China). pMD19-T vector, Hind restriction endonuclease III (Hind III), restriction endonuclease XbaI, and SYBR^®^ Premix Ex Taq^TM^ were purchased from Takara Co. (Shiga, Japan). pCMV-C-EGFP plasmid and pCMV-Blank plasmid were purchased from Tiangen Co. (Beijing, China). Lipofectamine^TM^ 2000 transfection reagent was purchased from Invitrogen Co. (Carlsbad, CA, USA). Sheep anti-rabbit TGR5 antibody, Rabbit anti-goat TGR5 antibody, and Rhodamine-labeled mouse anti-goat antibody were purchased from Cruz Santa Co. (Dallas, TX, USA). Horseradish peroxidase-labeled Goat anti-rabbit antibody was purchased from KPL Co. (Gaithersburg, MD, USA). cAMP ELISA Kit was purchased from Cayman Co. (Ann Arbor, MI, USA). The Bright-Glo^TM^ Luciferase Assay Kit was purchased from Promega Co. (Madison, WI, USA). The Nuclear Extract Kit was purchased from Active Motif Co. (Carlsbad, CA, USA). Dulbecco’s Modified Eagle Medium was purchased from Hyclone Co. (Shanghai, China). Fetal bovine serum was purchased from ExCell Co. (Taicang, China). Other reagents were purchased from ExCell Co.

### 2.2. Cell Lines

The 293T cells were purchased from the Shanghai Institute of Biochemistry and Cell Biology (SIBCB). Cells were cultured in 25 mL flasks and were maintained in DMEM with 10% FBS. Cells were cultured at a constant 37 °C under humidified conditions with 5% carbon dioxide (CO_2_). Non-adherent cells were discarded under the confluence of cells that reached approximately 70–80%. Adherent cells were digested with 0.25% trypsin, divided into three groups, and were incubated in the same medium. Cells were used in experiments from passages 3.

### 2.3. Construction of TGR5 Eukaryotic Expression Vector

Total RNA was extracted from human placenta tissue, and the cDNA was synthesized. The following TGR5 primers were used: F1: 5′-TCCCCAGGACCAAGATGACG-3′ and R1: 5′- GAGGCCCTTCCTTTAGTTCAAGTC-3′. The human TGR5 gene was connected to the pMD19-T vector and was transformed into competent cells. Then, the segment of pMD19-T-TGR5 was cloned into pCMV-Blank after sequencing, then the pCMV-TGR5 plasmid was obtained by Hind III and Xbal. The pMD19-T-TGR5 plasmid was the template, and the TGR5 target gene was cloned with the primers F2: 5′-CCCAAGCTTGGACCAAGATGACGCCCAA-3′ and R2: 5′-GCTCTAGAGTTCAAGTCCAGGTCGACA-3′. The pCMV-C-EGFP vector was constructed using two high-fidelity Hind III and Xbal polymerase, and the purified PCR products were connected with a pCMV-EGFP vector. Finally, the pCMV-EGFP-TGR5 plasmid was obtained after sequencing.

### 2.4. RNA Isolation and qPCR Assays

qPCR was adopted to quantify the mRNA level. The 293T cells were seeded in 24-well plates and were incubated overnight. The pCMV-TGR5 plasmid was transfected into the 293T cells using a time-dependent transfection method for 24 h. The total cellular mRNA was extracted using a Tri^TM^ RNA reagent. The quality of the mRNA was determined by agarose gel electrophoresis and the ratio of OD 260/280. cDNA was synthesized using the PrimeScript^TM^ RT Master Mix kit according to the manufacturer’s protocols. The cDNA was amplified using the SYBRs Premix-Ex-Tag^TM^ kit. In brief, a 25 μL reaction mixture was composed of 2 μL of cDNA, 12.5 μL of 2 × SYBRs Premix Ex Tag^TM^, 1 μL of specific target primers (10 μM), and 8.5 μL of ddH_2_O. The thermal cycling settings for qPCR were 30 s at 95 °C followed by 39 cycles for 5 s at 95 °C, 30 s at room temperature, and 15 s at 95 °C. of the qPCR primers that were used were glyceraldehyde-3-phosphate dehydrogenase (GAPDH)-F: 5′-GCACCGTCAAGGCTGAGAAC-3′ and GAPDH-R: 5′-TGGTGAAGACGCCAGTGGA-3′, and the TGR5 primers F: 5′-ATGACGCCCAACAGCA-3′ and R: 5′-AGGCAGGACCAGTAACCC-3′ were used.

All of the data were calculated based on the comparative Ct formula, and each sample was normalized to the GAPDH expression level. Relative mRNA expressions were analyzed according to the Ct values, which were based on the equation: 2^−ΔCt^ [ΔCt = Ct (TGR5) − Ct (GAPDH)]. 

### 2.5. TGR5 Receptor Localization

The 293T cells were cultured in 24-well plates with starvation treatment. The pCMV-EGFP-TGR5 plasmid was transiently transfected into the 293T cells using lipofectamine 2000. The culture medium was replaced with 10% DMED after 6 h. The expression of TGR5 was observed with an inverted fluorescence microscope after 24 h.

In order to observe the TGR5 receptor located in the membrane of the cells. The 293T cells were transfected with pCMV-EGFP-TGR5 for 24 h and were rinsed with phosphate-buffered saline (PBS) after the medium was discarded. The 293T cells were fixed in an immunostaining fixative for 15 min at room temperature and were washed with PBS three times (5 min each time). Then, these cells were permeabilized with 0.5% Triton X-100 for 10 min, washed with PBS three times (5 min each time), and blocked at room temperature for 1 h. Primary TGR5 antibodies at a 1:50 dilution were added and were incubated overnight at 4 °C. Three post-primary antibody washes were conducted, followed by the incubation of the 1:100 diluted secondary rhodamine-labeled fluorescence antibodies for 1 h at room temperature. Next, the 293T cells were washed and were subsequently stained with 4′,6-diamidino-2-phenylindole (DAPI) for 10 min at room temperature. DAPI was washed with PBS, and TGR5 expression was visualized with a confocal microscope (C2, Nikon).

### 2.6. Whole-Cell Protein Extraction

The whole-cell protein was extracted using a Nuclear Extract kit following the manufacturer’s protocols. In brief, the cells were washed with pre-cooled PBS/phosphatase inhibitor three times. Then, the supernatant was removed, and the pellet was re-suspended with 1 mL of PBS/phosphatase inhibitor and was centrifuged at 200× *g* and 4 °C for 5 min. After supernatant was discarded, 300 μL of complete lysis buffer was added into the samples and was incubated at 150 rpm for 20 min. The samples were vortexed and centrifuged at 14,000× *g* and 4 °C for 20 min. The supernatant was collected and stored at −80 °C.

### 2.7. Western Blot

The 293T cells were seeded in 6-well plates when cell fusion reached 70%, and the pCMV-TGR plasmid was transferred by means of transient transfection for 48 h. Total protein and concentration quantification were conducted according to the bicinchoninic acid assay (BCA) kit protocol. Equal amounts of protein (40–60 μg) were loaded into 8–12% SDS-PAGE gels and were transferred to polyvinylidene difluoride (PVDF) membranes. The TGR5 protein expression was determined using the primary TGR5 and GAPDH antibodies at 1:1000 and 1:3000 dilution. Horseradish peroxidase (HRP) conjugated secondary antibodies were used for band detection, and the bands were visualized using maximum sensitivity substrate. Chemiluminescence signals were measured using the enhanced chemiluminescence (ECL) system.

### 2.8. Internalization of TGR5 Receptor

The 293T cells were transiently transfected into the pCMV-EGFP-TGR5 cells. After 24 h, 10^−5^ M of TCDCA was added for 1 h. Then, the internalization of the TGR5 receptor was observed under an inverted fluorescence microscope.

### 2.9. TGR5 Luciferase Assay

The 293T cells were cultured in 24-well plates with DMEM medium containing 10% FBS. When cell fusion reached 70%, the culture medium was changed to DMEM. pGL4.29 and pCMV-TGR5 plasmids were co-transferred into the 293T cells by Lip2000. After 24 h of transfection, the cells were treated with TCDCA (100 μM, 10 μM, and 1 μM) and 100 μM of TLCA for 5 h, respectively. Finally, luciferase intensity was measured using a Bright-Glo^TM^ Luciferase Assay System.

### 2.10. cAMP Assay

The pCMV-TGR5 plasmid was transfected into the 293T cells and was incubated for 24 h. Then, 10–5 M of INT-777 treatment was implemented for 1 h. The rest of the 293T cell groups were treated with different concentrations of TCDCA and TLCA for 5 min, respectively. The cAMP ELISA Kit determined the intracellular cAMP.

### 2.11. Statistical Analysis

Each experiment was performed at least three times. All of the values are reported as mean ± S.D. between experimental groups and controls were assessed by one-way ANOVA with Dunnett’s test using SigmaStat (SPSS, Chicago, IL, USA). The level of statistical significance was set at *p* < 0.05.

## 3. Results

### 3.1. The Construction of pCMV-EGFP-TGR5 and PCMV-TGR5 Vectors

The TGR5 fragment was amplified from the human placenta tissues and was purified by PCR (Figure 1A,B). Then, the purified TGR5 fragment was cloned to a pMD19-T plasmid. The double enzyme digestion verified that the TGR5 was successfully targeted and inserted into the pMD19-T plasmid (Figure 1C). Continually, TGR5 targeted gene was cloned from pMD19-T-TGR5 plasmid, which acted as a pair of primers cloned to pCMV-EGFP plasmid tested by the double enzyme digestion and comparison on the GeneBank. Meanwhile, to verify whether the TGR5-targeted gene was successfully cloned into the pCMV-EGFP plasmid using PCMV-TGR5, the plasmid acted as a negative control. It turned out that both the pCMV-TGR5 and pCMV-EGFP-TGR5 plasmids were successfully constructed (Figure 1C–E).

### 3.2. TGR5 Receptor Expression in 293T Cells

The mRNA expression of TGR5 in the 293T cells that had been transfected with the pCMV-TGR5 plasmid was determined by qPCR. Meanwhile, TGR5 protein expression was analyzed with the TGR5 polyclonal antibody by means of Western blot. We found that TGR5 was significantly increased in 293T cells with the pCMV-TGR5 plasmid compared to the control. The gene and protein expression of TGR5 were consistent (Figure 2 and Figure 3).

To observe the expression and localization of TGR5 in the 293T cells that had been transfected by the pCMV-EGFP-TGR5 plasmid, the 293T cells were first incubated with the anti-TGR5 antibody and the secondary antibody labeled with rhodamine and DAPI. We found that the pCMV-EGFP-TGR5 plasmid was expressed in the 293T cells compared to in the negative control cells of the pCMV-EGFP plasmid using fluorescence confocal microscopy, confirming that TGR5 was expressed in the 293T cells (Figure 2A). Meanwhile, it was also observed that nuclei of the 293T cells were stained blue with the DAPI and TGR5 protein was green after it was combined with the green fluorescence. Moreover, the TGR5 protein appeared to be red after it was combined with the anti-TGR5 antibody and the secondary antibodies that had been labeled with rhodamine (Figure 3).

### 3.3. TCDCA Interacted with TGR5 Receptor

To directly observe the interaction between TCDCA and the TGR5 receptor, the pCMV-TGR5 and pCMV plasmids were transfected into 293T cells and with 100 μM of TCDCA treatment for 15 min, respectively. The invagination of the TGR5 receptor was observed using the inverted fluorescence microscope. The results indicated that the membrane fluorescence took place invagination in the 293T cells that had been treated with TCDCA for 15 min compared to the control group. The phenomenon also revealed that TCDCA interacted with the TGR5 receptor (Figure 4).

To further explore whether TCDCA interacts with the TGR5 receptor, 293T cells that were overexpressing TGR5 were treated with different concentrations of TCDCA (100 μM, 10 μM, and 1 μM) and 100 μM of TLCA for 1 h, with TLCA being used as a positive control. Meanwhile, IBMX acted as a TGR5-specific agonist, and DMEM acted as a negative control. A multi-function microplate reader determined that the TGR5 luciferase was a reporter gene. As a result, TCDCA and TLCA enhanced TGR5 luciferase reporter gene expression remarkably compared to the negative control. Moreover, the expression of the TGR5 luciferase reporter gene significantly increased in in the 293T cells that were overexpressing TGR5 and that had been treated with 100 μM of TCDCA than it was the 1 μM and 10 μM concentrations of TCDCA. TCDCA could significantly enhance the TGR5 luciferase reporter gene expression because TCDCA interacted with the TGR5 receptor (Figure 5).

The 293T cells were treated with different concentrations of TCDCA (100 μM, 10 μM, and 1 μM), 0.05 μM of INT-777, and 100 μM of TLCA for 1 h. The cAMP content was determined using an ELISA kit. It was determined that TCDCA and TLCA resulted in a remarkable increase in the cAMP content. Surprisingly, we also found that the cAMP content was augmented in the 293T cells that had been treated with 1 μM TCDCA, and it continued to increase in the 100 μM of TCDCA. Thus, these findings demonstrate that TCDCA enhances cAMP content in a concentration-dependent manner, indicating that TCDCA increases cAMP content through the TGR5 receptor (Figure 6).

## 4. Discussion

In this study, we investigated the interaction between TCDCA and TGR5. Our results indicated that TCDCA could activate the TGR5 receptor in 293T cells. Our previous research showed that TCDCA increased the apoptosis rate of NR8383 cells through the PKC-JNK signaling pathway [13]; we also showed that it was able to remarkably enhance the apoptosis level of adjuvant arthritis FLS in a dose-dependent manner [14]. Meanwhile, TCDCA decreased the expression of TNFα and IL-1β in rat serum treated with lipopolysaccharides (LPS) [13]. TGR5 is a receptor that is specific to bile acids and that belongs to the G-protein-coupled receptors (GPCRs). It is highly similar to other GPCRs in terms of both construction and characteristics. They may share some common effects in how they process mediating proteins. Until now, few papers that have reported on the interaction between TGR5 and bile acids. Here, we revealed that TCDCA interacted with TGR5, showing that it was able to activate it further and that it was also able to increase downstream cAMP content in the study. Throughout the entire process, TLCA was used as a positive control because TLCA, LCA, DCA, CDCA, and CA have been shown to regulate cAMP production by means of the TGR5 receptor in a dose-dependent manner in human TGR5-transfected Chinese hamster ovary (CHO) cells, but TLCA’s rank order of potency (EC50) is 0.33 μM, which is the highest compared to the other bile acids [15].

Many methods have been applied to aid in research focusing on the interaction between ligands and receptors in the biological field, such as radioactive labeling, fluorescent labeling, scintillation proximity assay (SPA), and fluorescence resonance energy transfer (FRET). However, it is challenging to label TCDCA using isotopes and fluorescence due to the fact that it is a particular structure [16,17,18,19]. Thus, we constructed a eukaryotic plasmid over-expressing TGR5, allowing us to continue investigating TGR5 localization, invagination, and cAMP content in 293T cells. Today, researching the interaction between ligands and receptors in biomedical science has become a popular trend. Numerous scientists are committing to selecting an effective and targeted receptor for pharmacological study. Therefore, our research has provided a unique method for demonstrating the interaction of the ligand and receptor.

The TGR5 receptor mediates a variety of signal pathways that are related to cancer, inflammation, diabetes, and obesity. Additionally, TGR5 receptor agonists significantly increase the cAMP content while also inhibiting downstream extracellular-signal-regulated kinase (ERK) signaling, inducing proliferation and inflammation in non-ciliated cholangiocytes [20]. Moreover, activated TGR5 receptors mediate the nuclear factor kappa light chain enhancer of activated B cells (NF-κB), regulating downstream transcript factors. In this way, the activated TGR5 receptor leads to several cellular processes, such as inflammation, proliferation, apoptosis, and development [21]. TGR5 receptors have many different agonists, but DCA and LCA are effective components that are able to inhibit tumor necrosis factor-α production in CD14^+^ macrophages by increasing the cAMP content to regulate NF-κB-p65 activation in the macrophages of liver cancer cells [22]. Glucagon-like peptide 1 (GLP-1), is able to increase insulin secretion after the oral administration of glucose due to the fact that it is an incretin hormone; however, TGR5 induces GLP-1 secretion via intracellular cAMP production [23]. Our research provides crucial evidence that supports the effects of TCDCA on diseases that function through the TGR5 receptor. Meanwhile, TGR5 can also applied to explore more bile acids and agonists for emerging diseases.

## 5. Conclusions

TCDCA acts on 293T cells, leading to TGR5 internalization as well as increases in cAMP and luciferase reporter. These results show that TCDCA can bind with the TGR5 receptor and can activate it, providing the experimental basis for the molecular mechanism of anti-inflammatory, immune regulation by TCDCA as well as a novel theoretical basis on which to seek a new target TCDCA.

## Figures and Tables

**Figure 1 molecules-26-07066-f001:**
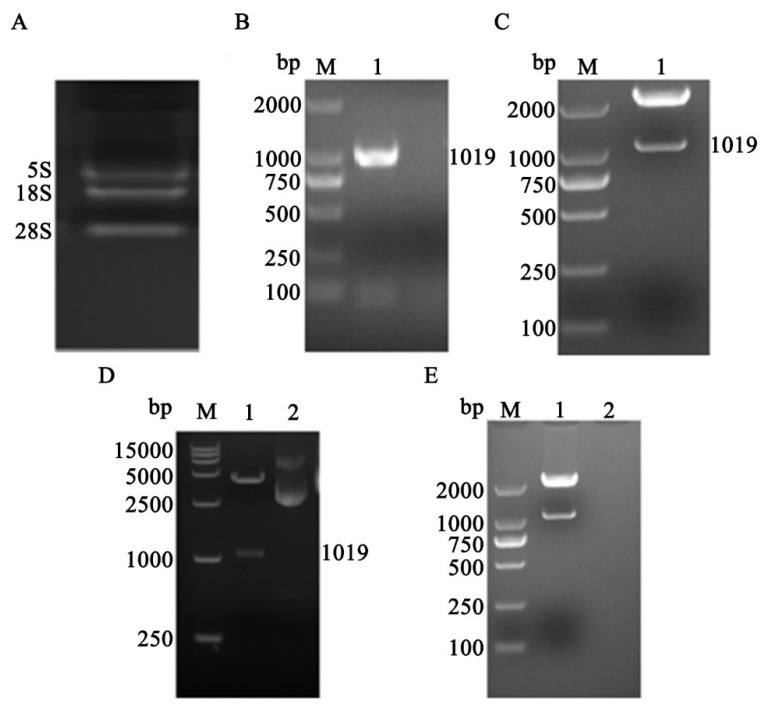
Construction of eukaryotic expression vector of pCMV-EGFP-TGR5 and pCMV-TGR5. (**A**): Total RNA extraction; (**B**): examination of TGR5; (**C**): the identification of pMD19-T-TGR5 by double enzyme cleavage; (**D**): the identification of PCMV-EGFP-TGR5 by double enzyme; (**E**): the identification of pCMV-TGR5 by double-enzyme cleavage.

**Figure 2 molecules-26-07066-f002:**
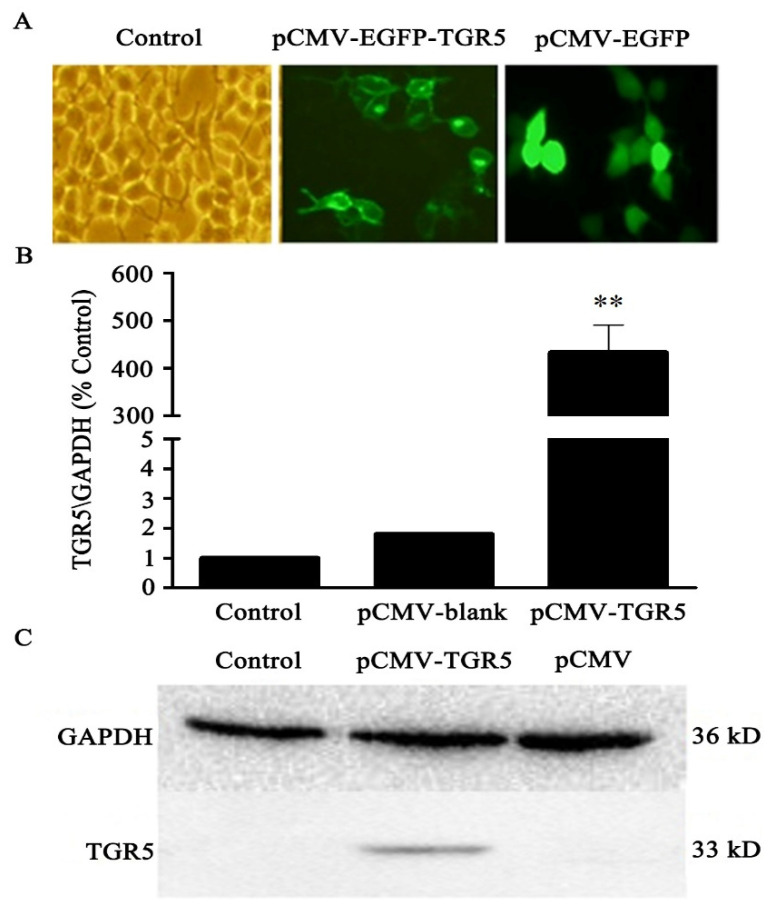
TGR5 mRNA and protein expression in 293T cells. (**A**): PCMV-EGFP-TGR5 labeled with the green fluorescent label can detect green fluorescence in 293T cell cells under a fluorescence microscope, which showed that PCMV-EGFP-TGR5 was expressed in 293T cells; (**B**): the 293T cells transfected with pCMV-TGR5 plasmid after TGR5 expression was increased compared to the control and pCMV-blank groups; (**C**): Western blot identified TGR5 recombinant protein. Data represents of means ± S.D. of three independent experiments. ** *p* < 0.01 vs. control.

**Figure 3 molecules-26-07066-f003:**
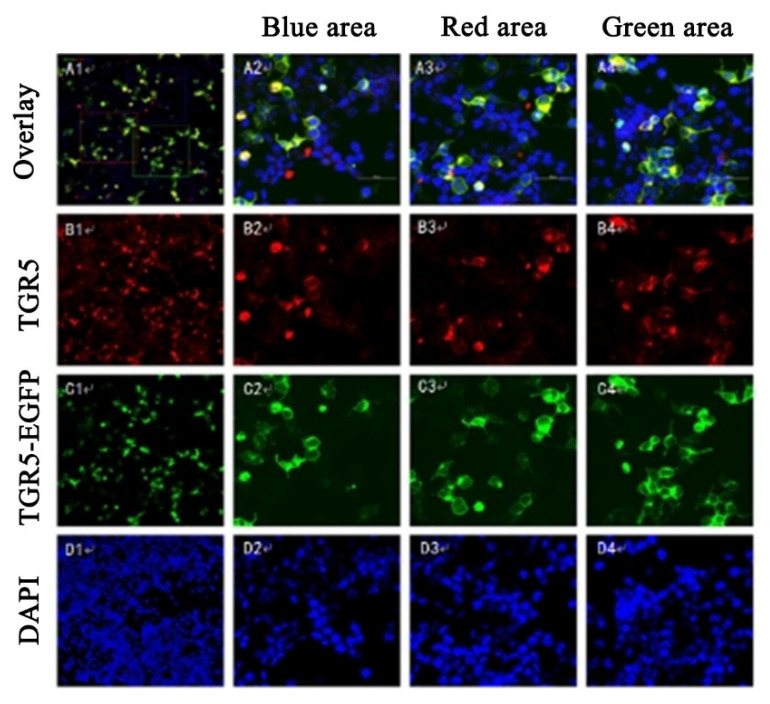
The expression and localization of TGR5 receptors were observed by immunofluorescence confocal microscopy. The red fluorescence overlay formation of yellow light distribution in the membrane and nuclei were stained blue with yellow around the blue, indicating receptor TGR5 in the 293T cells expressing it (**A1**–**A4**); TGR5 antibody-binding receptors present red fluorescence on the cell membrane (**B1**–**B4**); the 293T cells stably transfected with the pCMV-EGFP-TGR5 plasmid on the cell membrane (**C1**–**C4**); fluorescent dye DAPI and nuclear binding shown in blue (**D1**–**D4**).

**Figure 4 molecules-26-07066-f004:**
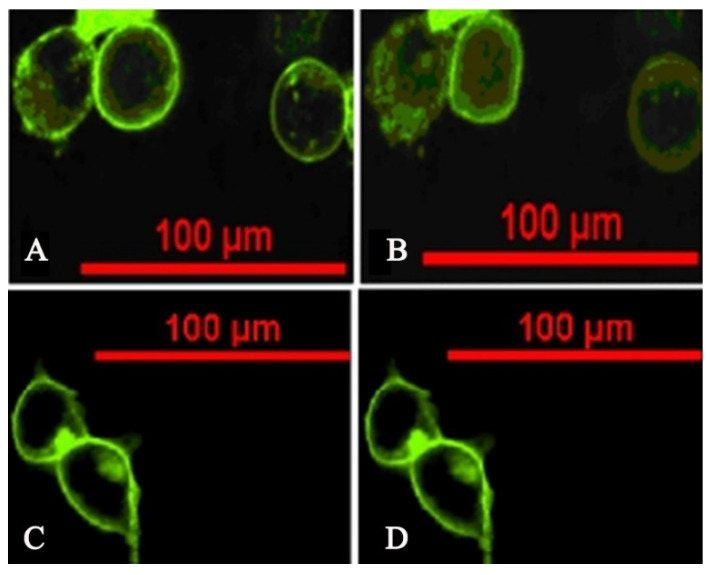
The invagination of TGR5. (**A**) transfected 293T cells treated with TCDCA for 0 min; (**B**) 293T cells treated with TCDCA for 15 min, which is when the fluorescence was observed; (**C**) untransfected 293T cells treated with TCDCA for 0 min; (**D**) untransfected 293T cells treated with TCDCA for 15 min.

**Figure 5 molecules-26-07066-f005:**
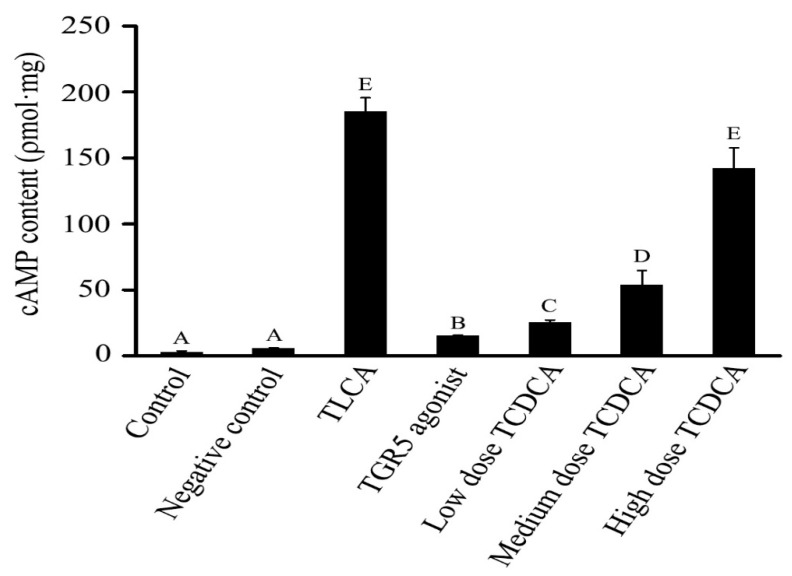
Luciferase reporter gene assay. TGR5 reporter gene was tested using the Bright-Gl^TM^ Luciferase Assay System. Negative control was untreated with TCDCA. Data represent of means ± S.D. of three independent experiments. The adjacent letters indicate significant differences vs. control (*p* < 0.05); the interphase letters indicate that the differences are extremely significant vs. control (*p* < 0.01).

**Figure 6 molecules-26-07066-f006:**
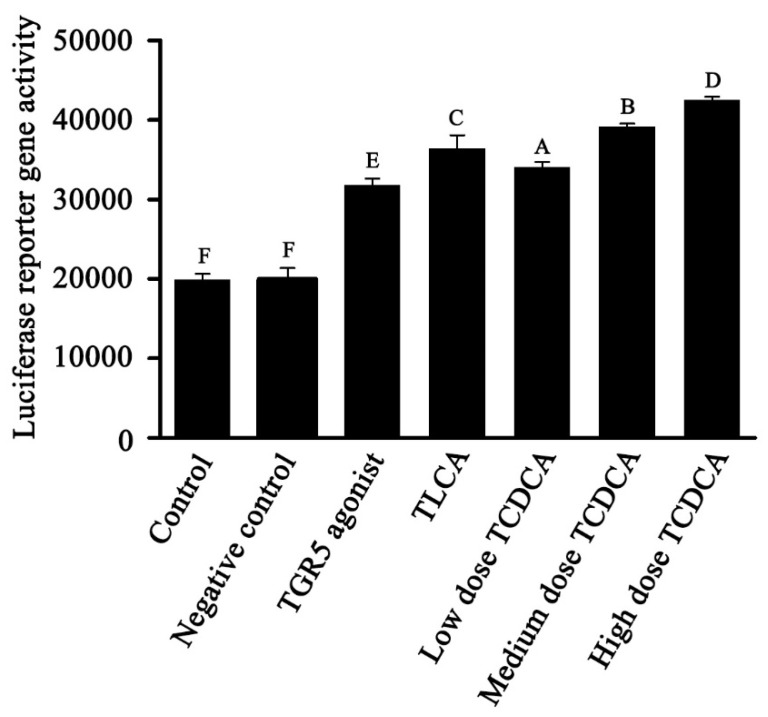
cAMP content detection. cAMP content assayed by ELISA. Negative control was not treated with TCDCA. The TLCA group was the positive control. Data represents means ± S.D. of three independent experiments. The adjacent letters indicate significant differences vs. control (*p* < 0.05); the interphase letters indicate that the differences are extremely significant vs. control (*p* < 0.01).

## Data Availability

The data used to support the findings of this study are available from the corresponding author upon request.

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
