# Peer review of "Taurochenodeoxycholic Acid Increases cAMP Content via Specially Interacting with Bile Acid Receptor TGR5"

_molecules, 2021, doi:10.3390/molecules26237066_

Round 1

Reviewer 1 Report

Interaction between Taurochenodeoxycholic acid and bile acid 2 receptor TGR5

Title

The title sounds more like a review article instead of original findings. I suggest to revise the title to fully reflect your manuscript content.

Abstract

  1. I suggest to rewrite this part ‘but no papers show that the effects have a relationship with TGR5’ to ‘ however it is not well known whether those effects are mediated by TGR5’. In my opinion, no papers word is a strong allegation.
  2. Rewrite ‘achieve the goal mentioned 18 above’ to ‘ achieve the aim’
  3. Not clear on this ‘Subsequent TGR5 expression, internalization, and downstream cAMP level were determined by 21 luciferase assay, fluorescence microscopy, and enzyme-linked immunosorbent assay (ELISA). The 22 results showed that 293T cells were successfully transformed with TGR5 expression’. As I think it is a bit redundant with this - the expression level in 293T cells was de- 19 termined by immunofluorescence, real-time quantitative PCR (RT-PCR, qPCR), and western blot. Did you meaning to say that luciferase assay, fluorescence microscopy, and enzyme-linked immunosorbent assay (ELISA) were done to check the interaction of TCDCA with TGR5?
  4. What about concentration of TCDCA used?

Introduction

  1. Line 33- what do you mean by the practical component?
  2. Line 37 – essential physiological role? – I suggest to replace with main or primary instead of essential.
  3. Line 40 – amphibian? Are you sure about the usage of the word here? Or amphipathic?
  4. Line 41- 44, please revise the sentence. I suggest to divide the long sentence into 2. In addition, in scientific writing usage of possessive pronoun for lab animals should be avoided.
  5. Line 49 – ‘following research of UDCA and others’- others referring to what?
  6. Line 51 – ‘a crucial apoptotic signal pathway’ – what is the crucial app signal pathway? Not clear
  7. Line 53- ‘the regular human hepatic cell lines’ what is this cell line?
  8. Line 57- typo ‘receptorm’
  9. Line 57-60 ‘6α-ethyl- (S)-methyl-cholic acid (EMCA, INT-777) was assessed for its agonistic activity on TGR5, activated TGR5 stimulates adenylate cyclase (AC), intracellular cAMP production and protein kinase A (PKA) activation in turn [12]’. I could not appreciate this sentence and link it with your reason why have done this work ‘so that our main aim is to elucidate an interaction between TCDCA and TGR5 receptor, a’.
  10. Line 61 – ‘a key therapeutic target for 61 TCDCA’- not clear. What therapeutic target? Your introduction was focusing on findings from previous work on TCDCA-pro apoptosis. Nothing about disease related issues of TCDCA. In my opinion, therapeutic is applicable to certain diseases or disorders.

Method

  1. Line 65-76 – is this a list or a sentences? I suggest to write in a more appropriate sentences.
  2. Cell lines – what are the passage used for this study.
  3. Line 86-87 ‘According to human TGR5 open reading frame sequence (NM_001077191.1) design primers of TGR5’ – please write in a more appropriate sentence, this comment should be applied for the manuscript not just this part. Past tense for methods.
  4. human placenta tissue was used, but not ethical approval number has been stated.
  5. ‘Expression of TGR5 was observed with an inverted fluorescence microscope after 24 h’ – is 24h enough for the TGR5 genes to be translated into a protein? Why 24 hr was chosen? Which work came first, immunofluorescence or the real time? I think it is more logical to check for gene expression and next should be immunofluorescence.
  6. Why was Nuclear Extract kit used for whole cell extraction?
  7. The method of internalisation detection in Line 152 was the same as in Line 99? Not clear
  8. ‘The rest of the 293T cells groups were treated with 164 different concentrations of TCDCA and TLCA for 5 min, respectively’ – please state the concentration used.

Result and discussion

  1. Result and discussion parts were poorly written. Difficulties to understand most of the sentences. Poorly constructed sentences with many grammatical errors.
  2. Why did you do both IF by confocal and fluorescence microscopy for the distribution?
  3. What do you mean by interacted? I am referring to your term – TCDCA interacted with TGR5
  4. Line 224 - what do you mean by ‘invagination’? is it the right term to be used here?
  5. Why TLC was used a control?
  6. How did you decide on the range of TCDCA concentration?
  7. What does the alphabet above the bar indicates? They were not significant? Not indicated by the *

Author Response

Dear reviewer,

The statements have been corrected. We will be happy to edit the text further besed on helpful comments from the reviewers. 

Sincerely,

Peifeng Li

Reviewer 2 Report

This study demonstrates interactions between TCDCA and TGR5.

Comments

1) Are the interactions specific for TGR5? Are there any interactions with FXRR?

2) Are the TGR5 antibodies specific and selective? Have the authors verified them in cells with TGR5 KO and FXR KO?

3) The studies should also be performed in polarized cells that are known to express TGR5.

4) Higher resolution immunofluorescence images of TGR5 are desirable to determine its precise subcellular localization at baseline and following stimulation with agonists.

Author Response

Dear Reviewer,

Thank you very much for your comments to our manuscript.

The statement have been corrented. We will be happy to edit the text further. Besed on helpful comments from the reviewer.

Best regards,

Peifeng Li

Round 2

Reviewer 1 Report

Authors have done extensive correction to improve the content of the manuscript according to my previous comments and suggestion. However, I still think the authors have not take into account my suggestion that extensive English editing is required.

These are a few examples grammar error:

  1. Line 27, shred? Is shred the right term to be used here?
  2. Title - the english does not sound right to me. My suggestion - TCDCA increases cAMP content upon interaction with the bile acid membrane receptor, TGR5
  3. Line 36, author starts the sentence with 'an main'. Authors need to get the manuscript proofread.
  4. Line 40, 'BAs have another function in the anti-inflammation and immunology' - anti inflammation is a part of immune responses.
  5. Line 73 - Lou Dang labeled horse antibody - what is Lou Dang?

I found similar mistakes almost in all pages. Thus, I suggest to get the manuscript proofread for language.
